# Facile synthesis of Fe-doped ZIF-8 and its adsorption of phosphate from water: Performance and mechanism

Zhijia Miao[1,2,3,4], Xueqiang Song[1,2], Xiaolei Wang[3], Hao Wang[4], Shuoyang Li[2,3], Zhen Jiao[1,2,3,4] *

1 Hebei Center for Ecological and Environmental Geology Research, Hebei GEO University, Shijiazhuang, China, 2 School of Water Resources and Environment, Hebei GEO University, Shijiazhuang, China, 3 Wastewater Treatment and Resource Reusing Technology Innovation Center of Hebei Province, Hebei Yuehai Water Group Co., Ltd., Shijiazhuang, China, 4 Norendar International Co., Ltd., Shijiazhuang, China

* Jiaozhen2012@126.com

**Data Availability Statement:** All relevant data are within the manuscript and its Supporting Information files.

## Abstract

To remove phosphate from water, a novel Fe-doped ZIF-8 was synthesized as a superior adsorbent. The Fe-doped ZIF-8 was fully characterized using different characterization techniques and it was found that the as-prepared Fe-doped ZIF-8 (denoted as ZIF-(2Zn:1Fe)) showed a polyhedral morphology with a large specific surface area of 157.64 $m^2$/g and an average pore size of 3.055 nm. Analyses using Fourier transform infrared (FTIR) spectroscopy and X-ray diffraction showed that Fe atoms were successfully incorporated into the ZIF-8 skeleton. Batch experiments demonstrated that the molar ratio of Fe and Zn has effects on phosphate adsorption. The adsorption kinetics conformed to a pseudo-second-order model with a high correlation coefficient ($R^2$ = 0.9983). The adsorption isotherm matched the Langmuir model ($R^2$ = 0.9994) better than the Freundlich model ($R^2$ = 0.7501), suggesting that the adsorption of phosphoric acid by ZIF-(2Zn:1Fe) can be classified as a chemisorption on a homogeneous surface. The adsorption amount was 38.60 mg/g. It was found that acidic environments favored the adsorption reaction and the best adsorption was achieved at an initial pH of 2. Inhibition of adsorption by common anions is $NO_3^- > CO_3^{2-} > SO_4^{2-} > Cl^-$. Characterization results indicate that the main mechanism of adsorption is surface complexation interactions.

## 1. Introduction

Discharge of industrial and municipal effluents containing elevated levels of phosphorus into natural water can lead to swift algae proliferation and eventually trigger water eutrophication [1]. This poses a threat to the survival of aquatic organisms and jeopardizes the quality of human drinking water [2]. Therefore, the development of effective phosphorus pollution management techniques becomes crucial [2–4]. Different techniques have been used to treat phosphorus in water, including biological, physical and chemical methods [5]. Among them, adsorption is one of the most effective technologies due to its simplicity, low cost and high

**Funding:** This study was financially supported by Hebei Provincial Natural Science Foundation, China in the form of a grant (C2021403002) received by ZM and ZJ. This study was also financially supported by the Open Project Program of Hebei Center for Ecological and Environmental Geology Research in the form of a grant (JSYF-202401) received by ZM and ZJ.

**Competing interests:** The authors declare that they have no known competing financial interests or personal relationships that could have appeared to influence the work reported in this paper.

efficiency [6–8]. Active carbon, mineral-based materials, ion exchange resins, metal oxides, layer double oxides, graphene-based composites, biomass, and their composites [9, 10] have been widely used as adsorbents for pollutants in water bodies. However, many adsorbents face various problems such as insufficient adsorption capacity, sluggish adsorption kinetics, limited adsorption selectivity, and insufficient regeneration capacity. To address these challenges, research efforts have focused on the development and synthesis of novel adsorbents to improve their adsorption performance for phosphorus [5].

Recently, metal-organic frameworks (MOFs) constructed from metals (metal clusters) and organic ligands have emerged as a porous crystalline material [11]. Their physicochemical properties, such as large specific surface area, ordered pore structures, easy of functionalization and designable properties, have attracted much attention in various fields [12]. To date, more than 20000 types of MOFs have been reported, primarily comprising of UiOs (discovered at the University of Oslo), MILs (discovered by the Materials Institute Lavoisier), PCNs (materials with a porous coordination network), and ZIFs (materials with a zeolitic imidazolate framework) [13].

ZIFs are heterogeneous materials with a porous structure analogous to that of zeolites, consisting of a connected network of four tetrahedral units in which metal ions such as $Zn^{2+}$ or $Co^{2+}$ are bridged by N atoms in the ectopic imidazolate anions [14]. With their large specific surface area, ZIFs have great potential for adsorption or catalytic conversion of pollutants in water [15–17]. For example, ZIF-8-modified $MnFe_2O_4$ exhibits excellent photo-Fenton catalytic degradation of tetracycline was reported, while Zn-O-Fe was demonstrated as the active site [18, 19]. In addition, the development of a heterogeneous structure of ZIF-8@carbon improves the ability to removal phosphate from water [20, 21]. In recent years, several studies have shown that metal doping and mixed ligands strategies can create unique structures of adsorbents [21, 22]. When two types of metal precursors are added to the system, both would participate in the reaction, resulting in the formation of a bimetallic skeleton, and the resulting heterostructure exhibits properties different from those of a single metal skeleton.

However, investigations on the synthesis, properties and phosphate adsorption of Fe-doped ZIF-8 have not yet been reported. Characterizing the structures of these materials and analyzing their adsorption properties for phosphate in water may help address the challenges of controlling water eutrophication.

In this study, several Fe-doped ZIF-8 adsorbents were synthesized by tuning the initial precursor mole ratios of Fe and Zn. The materials were fully characterized by different characterization tools. In batch adsorption experiments, the adsorption kinetics and isotherms of Fe-doped ZIF-8 on phosphates were studied. Furthermore, the effects of adsorbent dosage, initial solution pH (2.0–10.0), coexisting ions (chloride, sulfate, carbonate, and nitrate) and ionic strength on adsorption were evaluated.

## 2. Experimental section

### 2.1 Materials and reagents

Potassium dihydrogen phosphate ($KH_2PO_4$), Zinc nitrate hexahydrate ($Zn(NO_3)_2 \cdot 6H_2O$), ferric nitrate nonahydrate ($Fe(NO_3)_3 \cdot 9H_2O$), sodium chloride (NaCl), sodium nitrate ($NaNO_3$), sodium hydroxide (NaOH), sodium carbonate ($Na_2CO_3$), sodium sulfate ($Na_2SO_4$), and 2-methylimidazole (2-MI) were purchased from Sinopharm Chemical Reagent Co., Ltd. (Shanghai, China). Concentrated hydrochloric acid (HCl, 36–38%) and methanol was obtained from Aladdin reagent company (USA). All reagents were of analytical grade and used without further purification. Phosphate stock solutions were prepared by dissolving $KH_2PO_4$ in deionized water.

## 2.2 Synthesis of adsorbents

In a typical synthesis of Fe-doped ZIF-8, exactly, 0.2020 g Fe(NO$_3$)$_3$·9H$_2$O (0.5 mmol), and 0.2975 g Zn(NO$_3$)$_2$·6H$_2$O (1.0 mmol) were dissolved in 30.0 mL methanol (solution A). In addition, an amount of 0.8210 g 2-MI (10.0 mmol) were dissolved in 30 mL methanol (solution B). Then, Solution A was poured into solution B to form a homogeneous mixture, and the mixture was stirred well for 24 h at room temperature. At the end of the reaction, the sediment was centrifuged and collected and washed repeatedly with ultrapure water and anhydrous ethanol. Finally, the obtained product was dried in an oven at 80°C and left overnight for subsequent use and was recorded as ZIF-(2Zn:1Fe).

Using the same method, 0.2020 g of Fe(NO$_3$)$_3$·9H$_2$O was substituted with 0.4040 g (1.0 mmol) to produce ZIF-(1Zn:1Fe).

In the synthesis of ZIF-8, 0.2975 g of Zn(NO$_3$)$_2$·6H$_2$O (1.0 mmol) and 0.8210 g of 2-MI (10.0 mmol) were reacted to generate ZIF-8. In addition, the same synthetic method was used to prepare Fe-doped materials.

## 2.3 Characterization of materials

Powder X-ray diffraction (PXRD) patterns were recorded on a Panalytical X′Pert PRO diffractometer using monochromatized $K\alpha$ radiation ($\lambda$ = 0.154 nm). A Hitachi SU-8010 field emission scanning electron microscope (FESEM) equipped with an energy dispersive x-ray detector was used to study the microstructure and surface elements of the prepared adsorbents. Transmission electron microscope (TEM) images were taken with a Hitachi-7700 electron microscope. Fourier transform infrared (FT-IR) spectra were acquired on a Thermo Fisher Scientific Nicolet 6700 instrument. XPS was performed on a PHI 5000C ESCA spectrometer with a monochromatized Al $K\alpha$ source. The peaks were fitted with XPSPEAK41 software and the binding energy was corrected using the C 1s peaks as a reference (284.5 eV). Nitrogen adsorption-desorption isotherms were obtained at 77 K on a Quantum chromium Autosorb-1 apparatus.

## 2.4 Adsorption experiments

Adsorption experiments were carried out using the traditional bottle-point method to investigate the adsorption performance of the materials. Adsorption kinetics were determined by adding 5 mg of adsorbent into 10 mL solution with an initial concentration of phosphate (PO$_4^{3-}$) of 19.02 mg/L. Aliquots were extracted and filtered at predetermined time intervals. Similarly, the adsorption isotherm experiments were performed over a concentration range of 10.0–100.0 mg/L. The adsorption experiments were carried out for 24 h at 25°C on a shaker with 150 rpm to ensure the establishment of adsorption equilibrium.

The adsorption effect of adsorbent dosage in the range of 0.25 g/L ~ 1.00 g/L was evaluated. As well as the effect of initial solution pH in the range of 2.0 ~ 10.0 on adsorption was investigated. An amount of NaCl (20.0 mg/L) was added to the phosphate solution for ionic strength testing, respectively. To investigate the effect of coexisting anions on the experiments, stock solutions of interfering ions (0.1 mL, chloride (Cl$^-$), nitrate (NO$_3^-$), sulfate (SO$_4^{2-}$), and carbonate (CO$_3^{2-}$)) were added to 10 mL of phosphate stock solution to achieve a concentration of 10.0 mg/L of their respective anions. After adsorption, a small portion of the sample solution was filtered using a 0.22-μm syringe filter and analyzed by ammonium molybdate spectrophotometric at a wavelength of 700 nm on a UV-1780 spectrophotometer. The result was reported as PO$_4^{3-}$. In this study, two adsorption experiments were conducted and the mean values were reported. The quantity of adsorption ($q_t$, mg/g) and the amount of equilibrium adsorption ($q_e$, mg/g) were calculated using the adsorption equations (listed in S1 File) [23, 24].

## 3. Results and discussion

### 3.1 Structural analysis

The XRD patterns in Fig 1 featured two clear peaks below a 2θ value of 15° in the ZIF-(2Zn:1Fe) spectrum, which positions match well with those of the (011) and (112) planes of ZIF-8. The peaks at 35.1° and 62.5° belong to the peaks that are characteristic of Fe/Zn oxides or hydroxides. In the spectrum of ZIF-(1Zn:1Fe), no typical peaks of ZIF-8 were observed, while the two peaks at 35.1° and 62.5° strongly indicated the existence of oxides or hydroxides. The above results suggested that oxides or hydroxides were essential components of ZIF-(1Zn:1Fe), but only accounted for a small portion in ZIF-(2Zn:1Fe).

In the FT-IR spectra shown in Fig 2, the characteristic peak of Zn-N vibration was located at 419.4 $cm^{-1}$ of the ZIF-8 spectrum. However, the peaks of ZIF-(2Zn:1Fe) (441.6 $cm^{-1}$) and

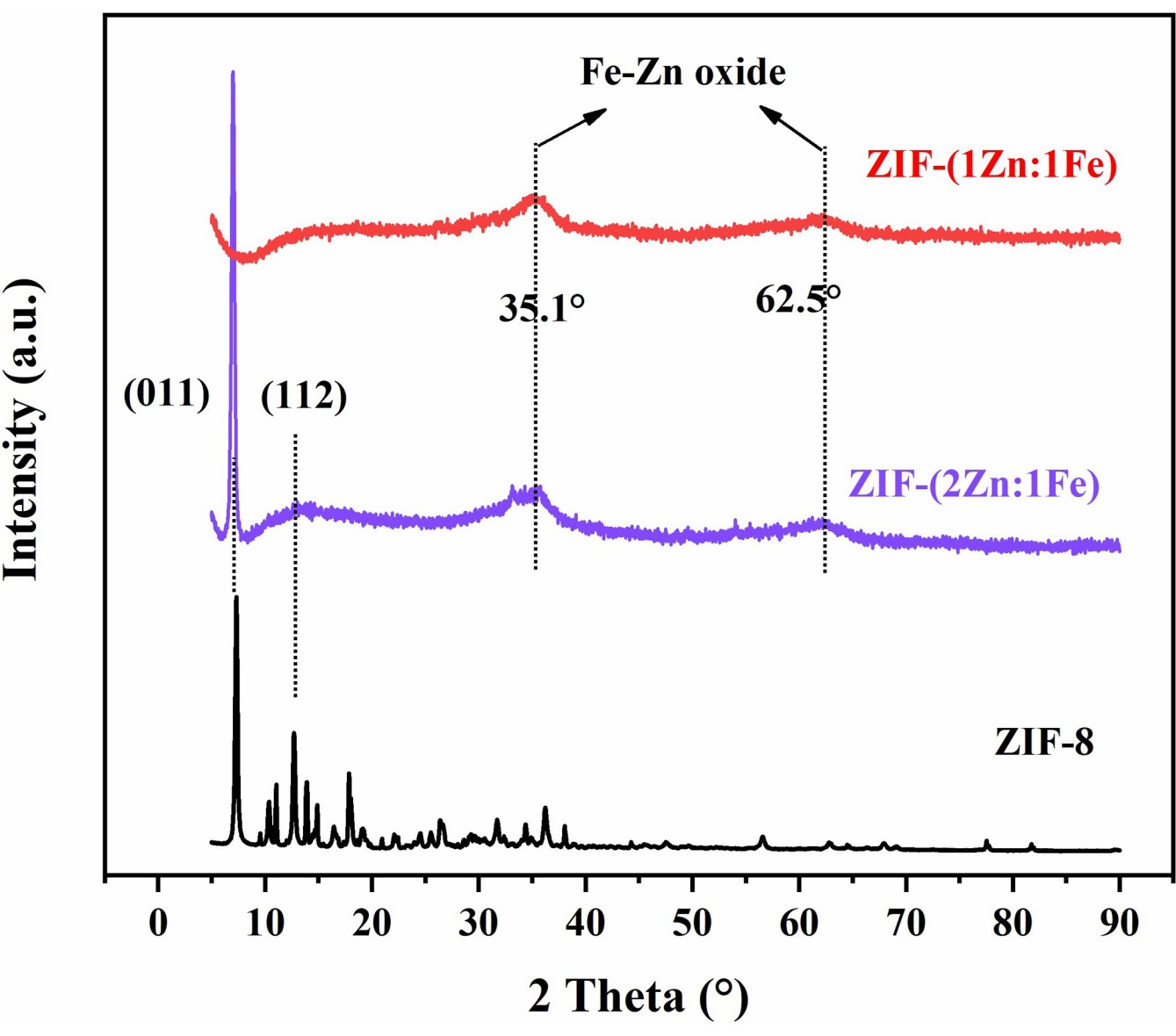

**Fig 1. XRD patterns of ZIF-8, ZIF-(2Zn:1Fe), and ZIF-(1Zn:1Fe).**

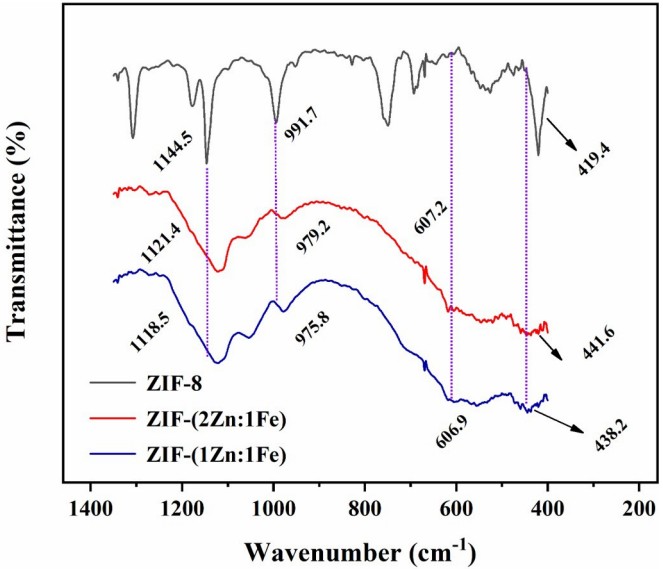

**Fig 2. FT-IR spectra of ZIF-8, ZIF-(2Zn:1Fe), and ZIF-(1Zn:1Fe).**

ZIF-(1Zn:1Fe) (438.2 cm$^{-1}$) are shifted to higher wave numbers. It can be inferred that the introduction of Fe atoms caused certain effects on the frameworks. The spectra of ZIF-(2Zn:1Fe) and ZIF-(1Zn:1Fe) show a new weak peak near 606 cm$^{-1}$ which was attributed to the stretching vibration of Fe-O, indicating that Fe atoms were doped into the framework of ZIF-8 to form a hybrid structure. Compared with ZIF-8, the intensity of all peaks associated with imidazole ring vibrations, such as the peaks at 980.0 cm$^{-1}$ and 1145.0 cm$^{-1}$, were severely weakened. These observations indicated that small amounts of oxides or hydroxides may be present in the iron-containing products, which agreed well with the SEM results.

SEM images of ZIF-(2Zn:1Fe) are shown in Fig 3. The low magnification images showed that ZIF-(2Zn:1Fe) appeared in different sizes clumps, which could be attributed to the agglomeration behavior of nanostructures (Fig 3A and 3B). At high magnifications, a clear microstructure consisted of many small particles were visible in the SEM images. These particles were homogeneous and had sizes in the order of tens of nanometers (e.g., the orange circled zone in Fig 3C and 3D). This structure was similar with the one previously reported in literature [14]. When the mole ratio of Zn and Fe changed from two to one, the hybrid frameworks were not apparent. The corresponding metal oxides or hydroxides appear to be produced in an alkaline environment (from 2-MI). Compared with ZIF-(1Zn:1Fe), ZIF-(2Zn:1Fe) may exhibit different adsorption properties.

In the full XPS spectrum of ZIF-(2Zn:1Fe) in Fig 4, the peaks of Zn 2p, Fe 2p, N 1s, and C 1s could be clearly identified, indicating that ZIF-(2Zn:1Fe) mainly composed of the four elements, namely, zinc, iron, nitrogen and carbon. In combination with other characterization results, it can be inferred that Fe atoms have been successfully introduced into the framework of ZIF-8, and the resulting product may contain small amounts of oxides and/or hydroxides. According to the literature, the combination of MOFs and oxides in the as-prepared composites may have important effects on adsorption performance.

The nitrogen adsorption-desorption isotherm of ZIF-(2Zn:1Fe) and the corresponding pore size distribution curve are shown in Fig 5. The adsorption/desorption isotherm (Fig 5A) is a type II isotherm with an H3 hysteresis loop. The BET surface specific area of ZIF-(2Zn:1Fe) was calculated using the BJH model (desorption branch) to be 157.64 m2/g, with a

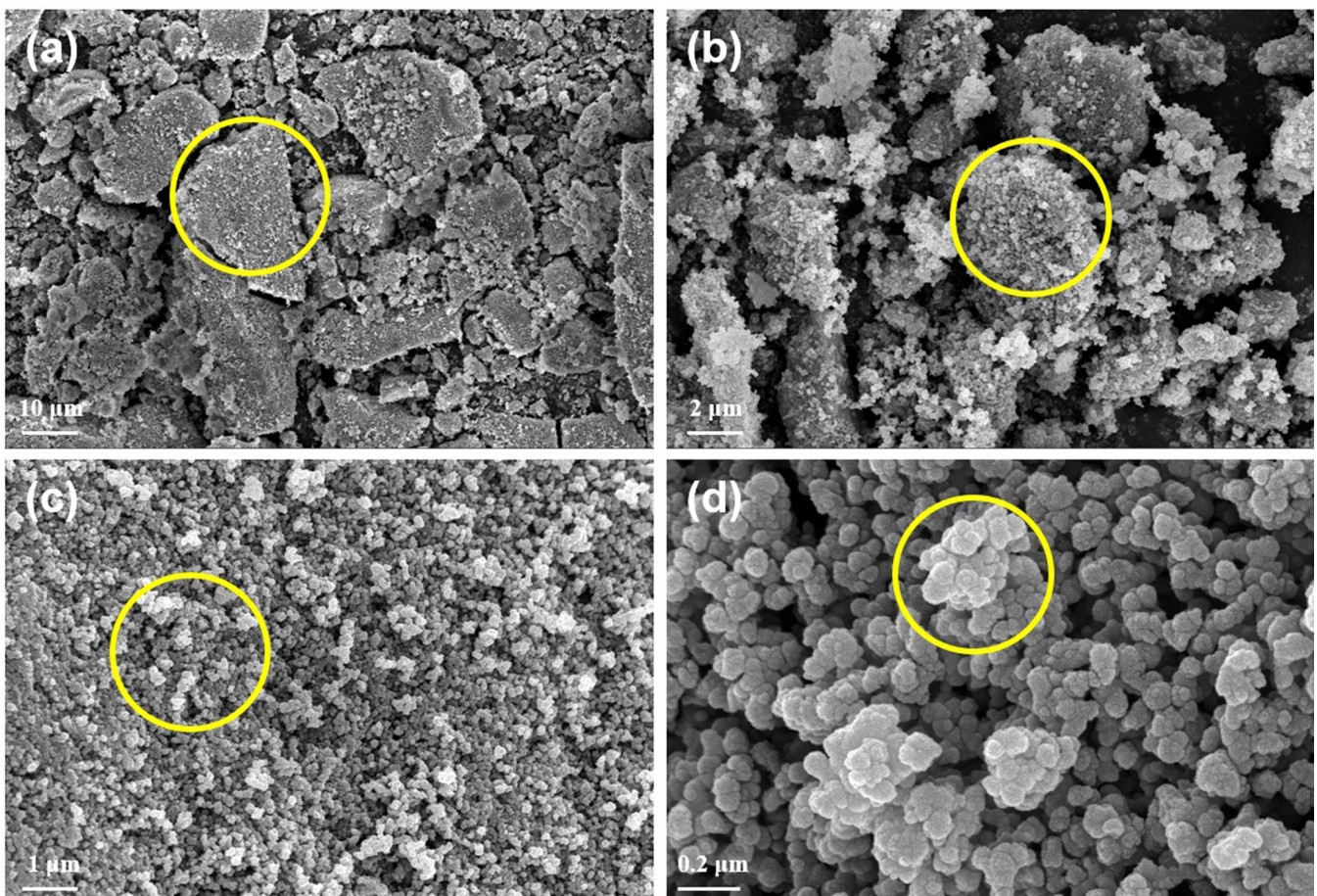

**Fig 3. SEM images of ZIF-(2Zn:1Fe) at different scales.**

pore volume of 0.285 cm$^3$/g and a pore size of 3.055 nm (Fig 5B). This mesoporous material could facilitate adsorbate diffusion through the pores and accelerate adsorption kinetics.

### 3.2 Adsorption experiment

To evaluate the adsorption performance of ZIF-(2Zn:1Fe), ZIF-(1Zn:1Fe) and ZIF-8, the phosphate concentration $C_t$ was calculated for each system and compared with the contact time ($t$) as shown in Fig 6A. The $C_t$ values decreased rapidly to low levels in all three systems. The $C_t$ values of ZIF-8, ZIF-(2Zn:1Fe) and ZIF-(1Zn:1Fe) were reduced to 6.67, 2.63, and 6.25 mg/L, respectively, indicating that ZIF-(2Zn:1Fe) was the most effective material for adsorption of phosphates. An excess of Fe atoms inhibits the formation of MOFs resulting in a very small specific surface area with only a few adsorption sites exposed, whereas ZIF-(2Zn:1Fe) maintains the mesoporous structure of ZIF-8 with Fe doping, thus enhancing the performance compared to ZIF-8. In Fig 6B, the phosphate adsorption $q_t$ was varied with contact time ($t$). It is obvious that the of ZIF-8, ZIF-(2Zn:1Fe) and ZIF-(1Zn:1Fe) increased immediately from 0 to 12.36, 16.39 and 12.77 mg/g, respectively, at the first measurement, where the highest adsorption was observed for ZIF-(2Zn:1Fe). Among them, ZIF-(2Zn:1Fe) showed the highest adsorption amount. The adsorption of ZIF-(2Zn:1Fe) was increased by 32.60% compared to ZIF-8 due to Fe doping. However, excessive Fe atoms inhibited the formation of the MOFs,

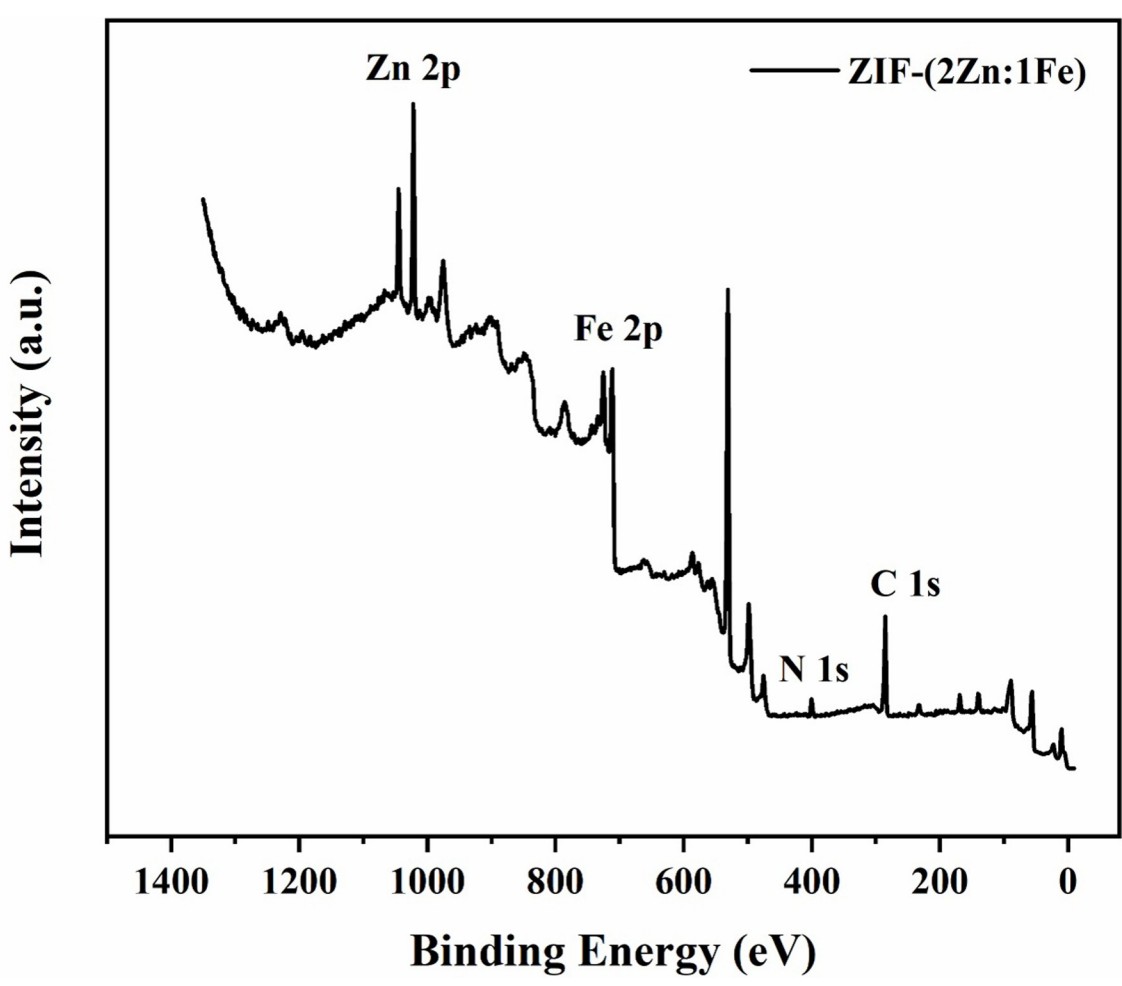

**Fig 4. Full XPS spectra of ZIF-(2Zn:1Fe).**

which leads to a decrease in the adsorption amount. Consequently, ZIF-(2Zn:1Fe) was an optimal adsorbent.

**3.2.1 Adsorption kinetics.** To investigate the mass transfer mechanism, two typical kinetic models, i.e., the pseudo-first-order model (PFO) and pseudo-second-order model (PSO), were used to simulate the experimental data using the following equations (listed as follows) [25–27].

The linear curves are shown in Fig 6A and 6B, and their corresponding fitting parameters are listed in Table 1. The PSO coefficient of determination, $R^2$, for ZIF-8, ZIF-(2Zn:1Fe), and ZIF-(1Zn:1Fe) were 0.9983, 0.9983, and 0.9968, respectively. These values were much higher than the PFO and were all lower than 0.85. In addition, the calculated values of the first $q_t$ (12.64, 16.75, and 13.12 mg/g) were much closer to the experimental values (12.36, 16.39, and 12.77 mg/g). This suggested that the adsorption kinetics of phosphate were in good agreement with the PSO, indicating that chemical adsorption as the rate-limiting step [28, 29]. The kinetic constants of the PSO fitted equations were 0.1793, 0.1479, and 0.5426 g/(mg h), respectively, which further suggested that ZIF-(2Zn:1Fe) was an optimal adsorbent with the fastest adsorption kinetics (data listed in Table 1). This performance was largely dependent on the unique mesoporous structure of ZIF-(2Zn:1Fe) [10].

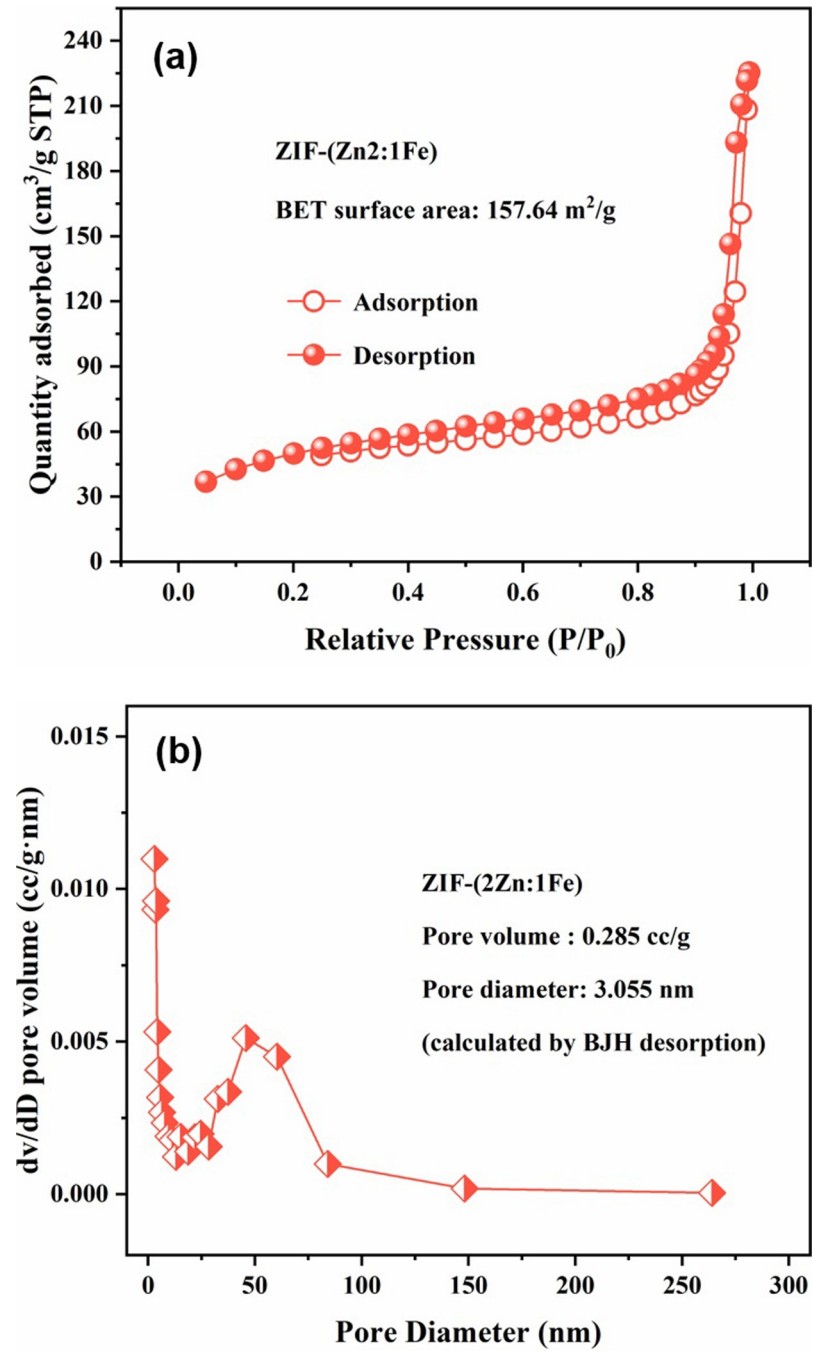

**Fig 5.** Nitrogen adsorption-desorption isotherm (a) and pore size distribution (b) of ZIF-(2Zn:1Fe).

**3.2.2 Adsorption isotherm.** To explore the adsorption mechanism, isotherm adsorption experiments were carried out at 25˚C using ZIF-(2Zn:1Fe) and different initial phosphate concentrations. In Fig 7A, the equilibrium adsorption amount ($q_e$) was related to the natural logarithm of the equilibrium concentration ($C_e$) after 24 h of exposure. After a rapid increase, the growth rate of $q_e$ slowed down and reached a plateau at higher concentrations of phosphate.

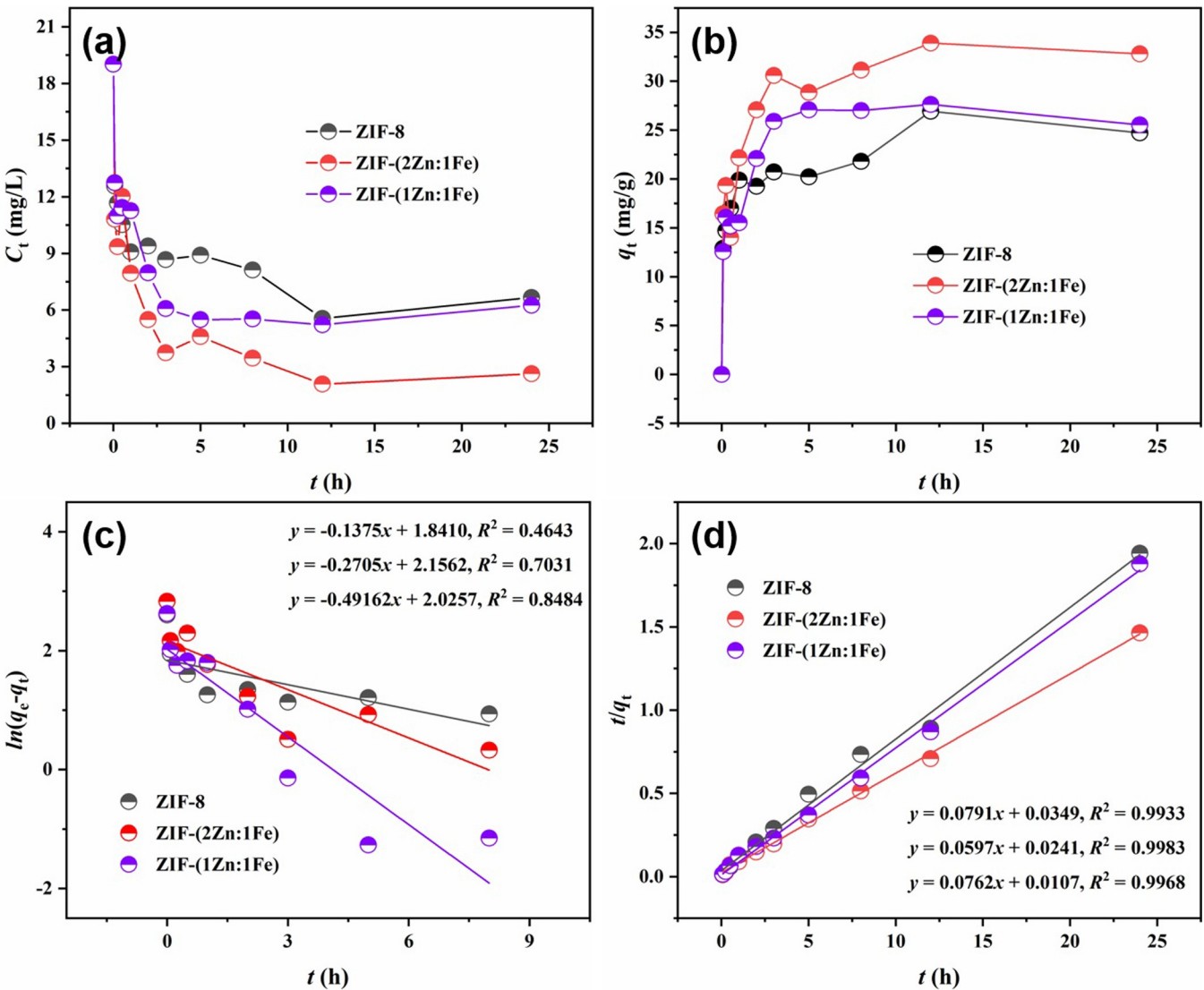

**Fig 6.** Effects of contact time on phosphate concentration in solution (a) and adsorption amount (b); pseudo-first-order model (c) and pseudo-second-order model (d). Experimental conditions: $[PO_4^{3-}]0 = 19.02$ mg/L, dosage: 0.5 g/L, T = 298 K, pH = 7.0, t = 24 h.

The maximum adsorption amount was 38.60 mg/g when $C_e$ was 19.78 mg/L. In theory, the active sites of ZIF-(2Zn:1Fe) were completely occupied under this condition.

Two known isothermal models, the Langmuir model and the Freundlich model, were used to fit the experimental data, and the fitted linear equations were listed as follows [30, 31].

**Table 1. Kinetics parameters of two kinetic models (Fig 6).**

| Adsorbents | Pseudo first-order model | | | | Pseudo second-order model | | | |
|---|---|---|---|---|---|---|---|---|
| | $R^2$ | $q_e$ (mg/g) | $k_1$ (1/h) | RSS | $R^2$ | $q_e$ (mg/g) | $k_2$ (g/(mg h)) | RSS |
| ZIF-8 | 0.4643 | 6.303 | 0.1375 | 0.9942 | 0.9933 | 12.64 | 0.1793 | 0.0193 |
| ZIF-(2Zn:1Fe) | 0.7031 | 8.638 | 0.2705 | 1.5308 | 0.9983 | 16.75 | 0.1479 | 0.0028 |
| ZIF-(1Zn:1Fe) | 0.8484 | 7.581 | 0.4916 | 2.2036 | 0.9968 | 13.12 | 0.5426 | 0.0084 |

Note: RSS means residual sum of squares.

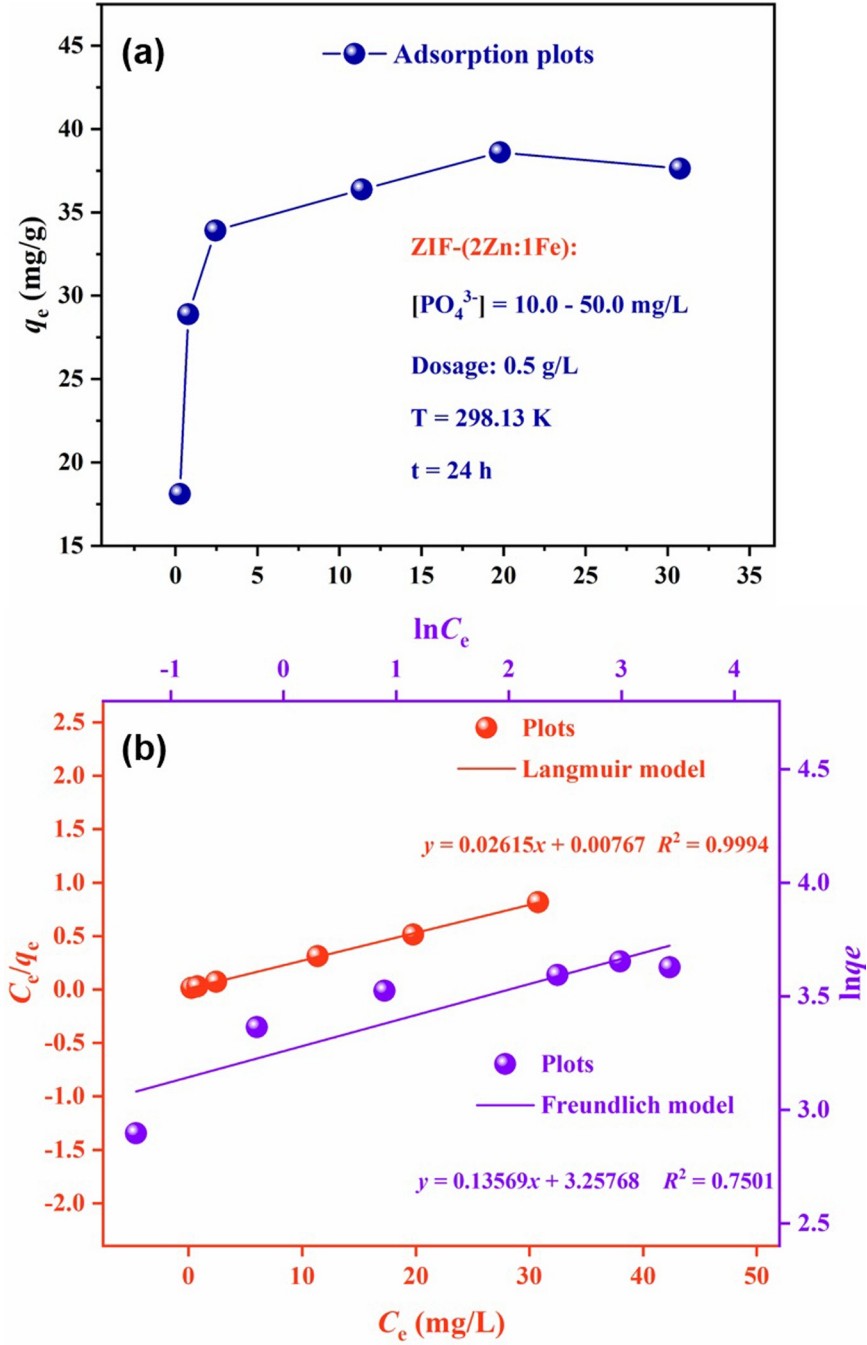

**Fig 7.** Adsorption isotherm of ZIF-(2Zn:1Fe) (a) and Langmuir model and Freundlich model (b). Experimental conditions: $[PO_4^{3-}]_0$ = 10.0–50.0 mg/L, dosage: 0.5 g/L, T = 298 K, pH = 7.0, t = 24 h.

The fitted curves are shown in Fig 7B, and the corresponding parameters are listed in Table 2. The $R^2$ values of the Langmuir model and Freundlich models were 0.9994 and 0.7501, respectively. Therefore, the Langmuir model was more suitable for describing the adsorption process than the Freundlich model. The results suggested that the adsorption of phosphate by ZIF-(2Zn:1Fe) was chemisorbed on a homogeneous surface [32, 33]. The adsorption capacity of ZIF-(2Zn:1Fe) was as high as 38.24 mg/g, which was superior to most of the comparable

**Table 2. Isotherm parameters of two isothermal models (Fig 7).**

| Langmuir model | | | | Freundlich model | | | |
|---|---|---|---|---|---|---|---|
| $R^2$ | $q_m$ (mg/g) | $K_L$ (L/mg) | RSS | $R^2$ | $n$ | $K_F$ (mg g$^{-1}$) (L mg$^{-1}$)$^{1/n}$ | RSS |
| 0.9994 | 38.24 | 3.41 | 0.0002 | 0.7501 | 7.37 | 25.99 | 0.0826 |

Note: RSS means residual sum of squares.

adsorbents (Table 3) [34–37]. Based on these comparisons, this material had great potential for practical applications.

**3.2.3 Effects of environmental conditions.** *Effects of dosage.* In general, the adsorbent dosage affects the adsorption performance, and the dosage is directly correlated with the number of active sites. As shown in Fig 8A, the adsorption of ZIF-(2Zn:1Fe) decreased gradually from 36.10 mg/g to 20.29 mg/g, when the dosage was incrementally increased from 0.25 to 1.00 g/L. From the economical point of view, a proper adsorbent dosage (0.5 g/L) was selected for the subsequent experiments. Because this dosage ensured a high adsorption capacity and removal efficiency.

*Effects of initial solution pH.* The pH of the solution was an important factor as it can change the surface charge of the adsorbent and the type of adsorbent available. As shown in Fig 8B, the maximum adsorption of ZIF-(2Zn:1Fe) was 35.03 mg/g at pH of 2.0, which gradually with increasing pH. The adsorption amounts were 29.16 and 29.30 mg/g in weakly acidic and neutral environments (pH 4.0 or 6.0), respectively, and close to 20 mg/g in alkaline environments (pH 8.0 or 10.0). This phenomenon was explained by the fact that at a pH of 2.0, phosphate existed mainly in the form of $H_3PO_4$ and $H_2PO_4^-$, which showed strongly and complexly interact with active sites of ZIF-(2Zn:1Fe). Two or three hydroxyl groups in each molecule could react with the surface of the adsorbent to produce a large adsorption capacity. At pH of 4.0 or 6.0, phosphate existed in either $H_2PO_4^-$ or $HPO_4^{2-}$ form, whereas at pH of 8.0 or 10.0, phosphate existed almost exclusively as $HPO_4^{2-}$ [2, 38]. The affinity was strongly correlated with the number of hydroxyl groups, which explains the observed phenomenon that the fewer the hydroxyl groups, the lower the adsorption at elevated pH. Therefore, the adsorption of phosphate must occur through complexation reactions [28].

*Effects of coexisting ions.* Many anions are present in natural water and they usually interact with phosphate. It was essential to study their effects on phosphate adsorption from water. As shown in Fig 8C, $Cl^-$ and $SO_4^{2-}$ had a slight inhibitory effect on phosphate removal, with phosphate adsorption decreasing from 31.16 mg/g to 27.50 mg/g and 28.88 mg/g, respectively. $CO_3^{2-}$ exerted a moderate effect on phosphate adsorption with a reduction of 10.14 mg/g, while $NO_3^-$ exerted a severe inhibitory effect with a reduction of 14.81 mg/g, which was almost 50% compared to the control group. Due to the similar nature of $NO_3^-$ shared similar properties with $H_xPO_4^{3-x}$, these two anions strongly competed for the same active sites of ZIF-(2Zn:1Fe). Previous studies had shown that $CO_3^{2-}$ could be adsorbed on adsorbent surface

**Table 3. Comparison of $PO_4^{3-}$ adsorption capacity with reported adsorbents.**

| Adsorbent | Concentration (mg/L) | Maximum capacity (mg/g) | References |
|---|---|---|---|
| PVA-CSH | 5.0–120 | 28.15 | (Ding et al, 2018) |
| MCM-41 | 90–450 | 21.01 | (Seliem et al, 2016) |
| CDC | 1.0–100 | 16.14 | (Almanassra et al, 2020) |
| PS-M–LDH | 5.0–50 | 34.20 | (Feng et al, 2022) |
| ZIF-(2Zn:1Fe) | 5.0–50.0 | 38.24 | This study |

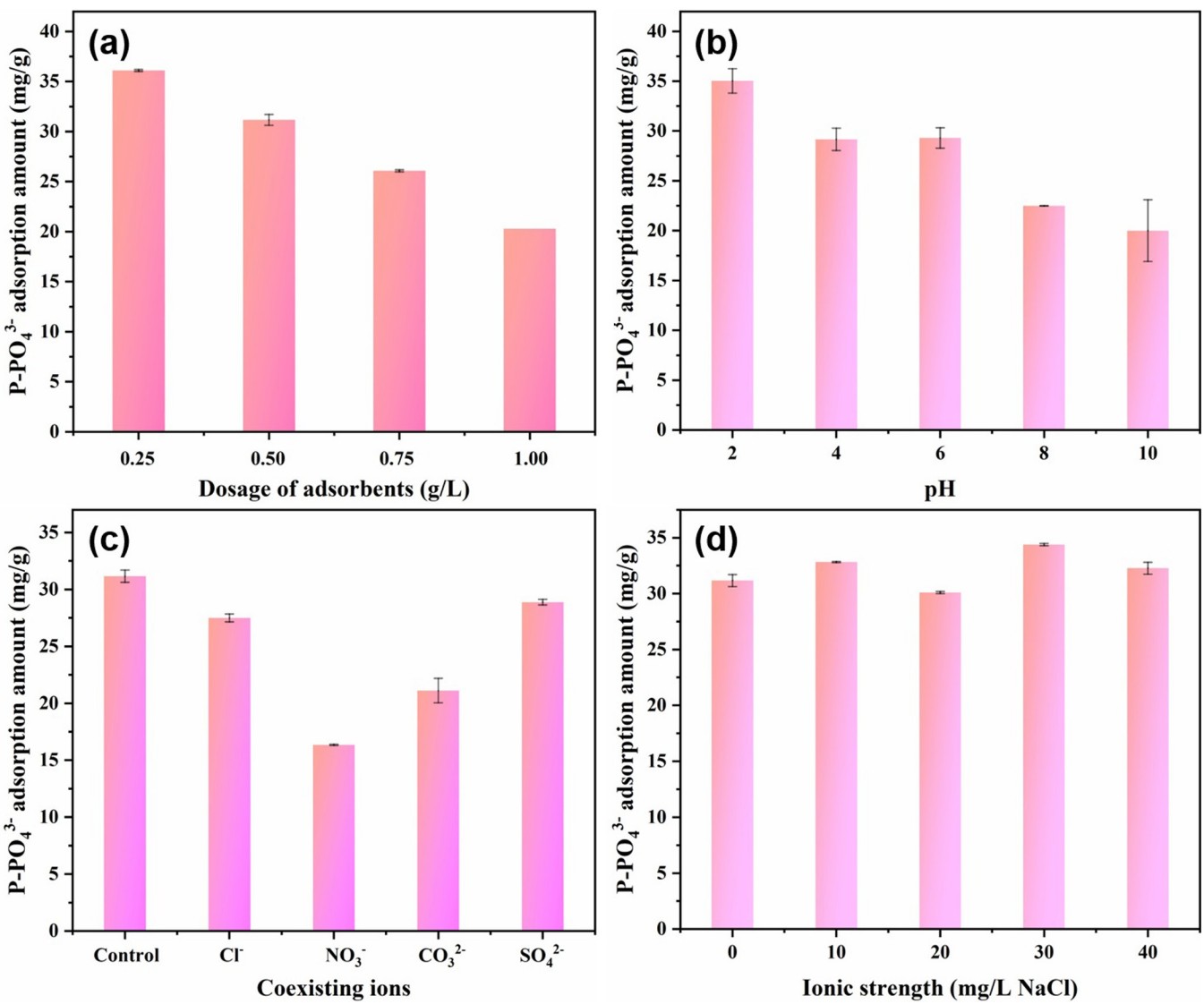

**Fig 8.** Effects of adsorbent dosage (a), initial solution pH (b), coexisting ions (c), and ionic strength (b) on $PO_4^{3-}$ adsorption. Experimental conditions: $[PO_4^{3-}]$ 0 = 20.0 mg/L, dosage: 0.5 g/L, T = 298 K, pH = 7.0, t = 24 h.

through the formation of outer or inner sphere complexes [39]. Phosphate was mainly adsorbed through the formation of inner-sphere complexes, while $CO_3^{2-}$ may partially interfere with phosphate adsorption [40].

*Effects of ionic strength*. As shown in Fig 8D, phosphate adsorption only fluctuated negligibly within a small range of 0 to 4.29 mg/g as the ionic strength was increased from 0 to 40 mg/L (NaCl). This slight effect suggested that ionic strength was not the limiting factor for phosphate removal. As a matter of fact, the inner-sphere complexes were not directly affected by the ionic strength. Consequently, this observation confirmed that phosphate should be removed from solution through the formation of inner-sphere complexes [41].

### 3.3 Adsorption mechanism

In this study, phosphate adsorption mechanisms were systematically explored by SEM observations, SEM-EDX, FT-IR spectroscopy, XRD, and XPS. The specific results are shown in Figs 9–11.

As can be seen from the SEM image Fig 9A, the microstructure of ZIF-(2Zn:1Fe) did not undergo any significant changes after adsorption. Five sites of the used adsorbents were detected and analyzed using SEM-EDX technique (as shown in Fig 9B). The signal of the P element appeared in a representative spectrum (Fig 9C). The elemental composition of the five sites were also tabulated (Fig 9D), showing an average atomic percentage of P of 21.10%. These analyses indicated that the phosphate was successfully attached to the adsorbent, which was consistent with the results of the adsorption analysis.

In the FT-IR spectrum of the used adsorbent (Fig 10A), a new peak appeared near 580 cm$^{-1}$, which should represent the vibration of $PO_4^{3-}$. The XRD pattern of the ZIF-(2Zn:1Fe) demonstrated that the material changed significantly after adsorption after use (Fig 10B). After adsorption, the peak of the (011) lattice facet of ZIF-(2Zn:1Fe) almost disappeared, which implies that a strong adsorption reaction took place or that the lattice face may contain important active sites, which is consistent with previous findings that Fe doping into the frameworks produced some active sites [28, 38]. Therefore, XPS spectra were further analyzed to identify the active sites.

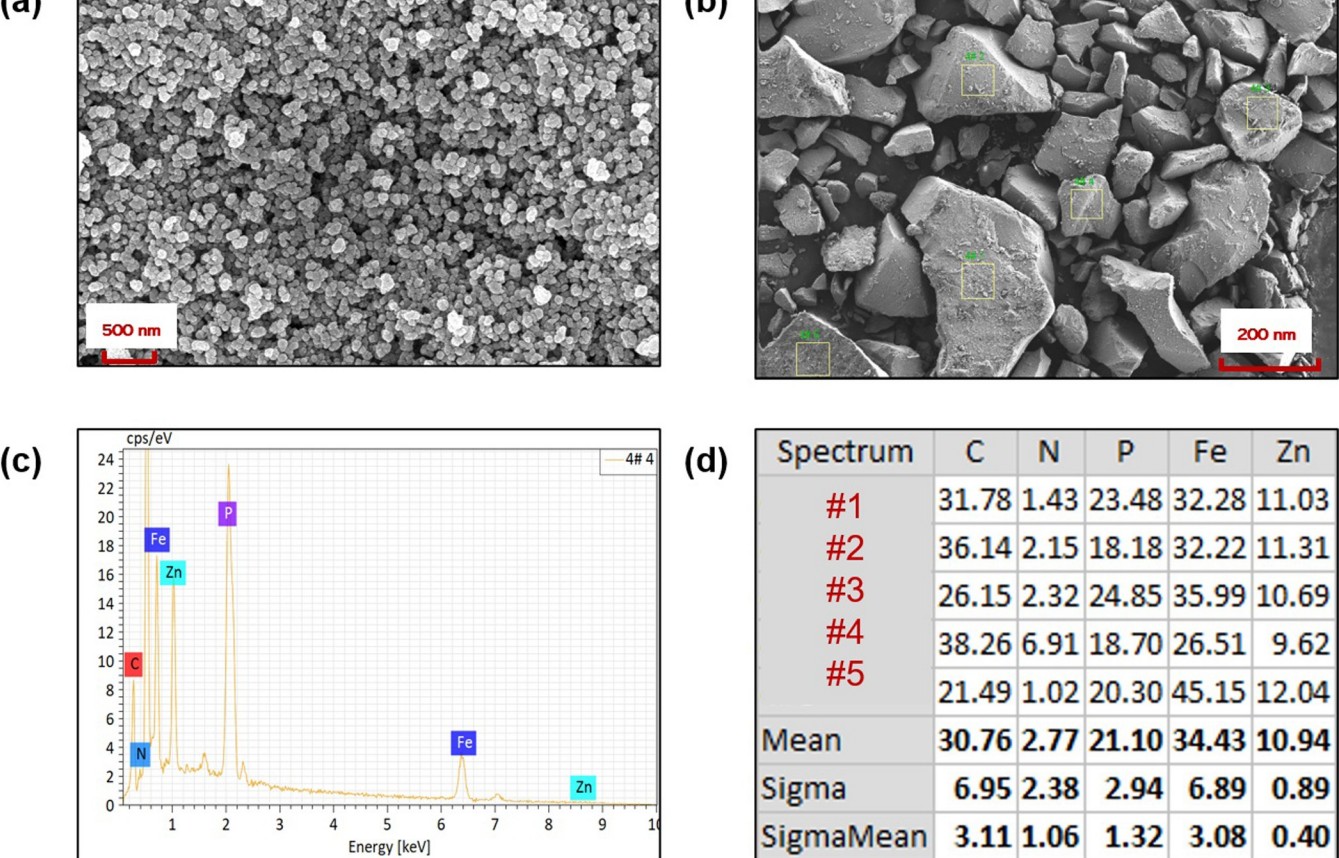

**Fig 9.** SEM images (a, b), EDX spectra (c), and atomic fractions of ZIF-(2Zn:1Fe)- $PO_4^{3-}$ as percentage (d).

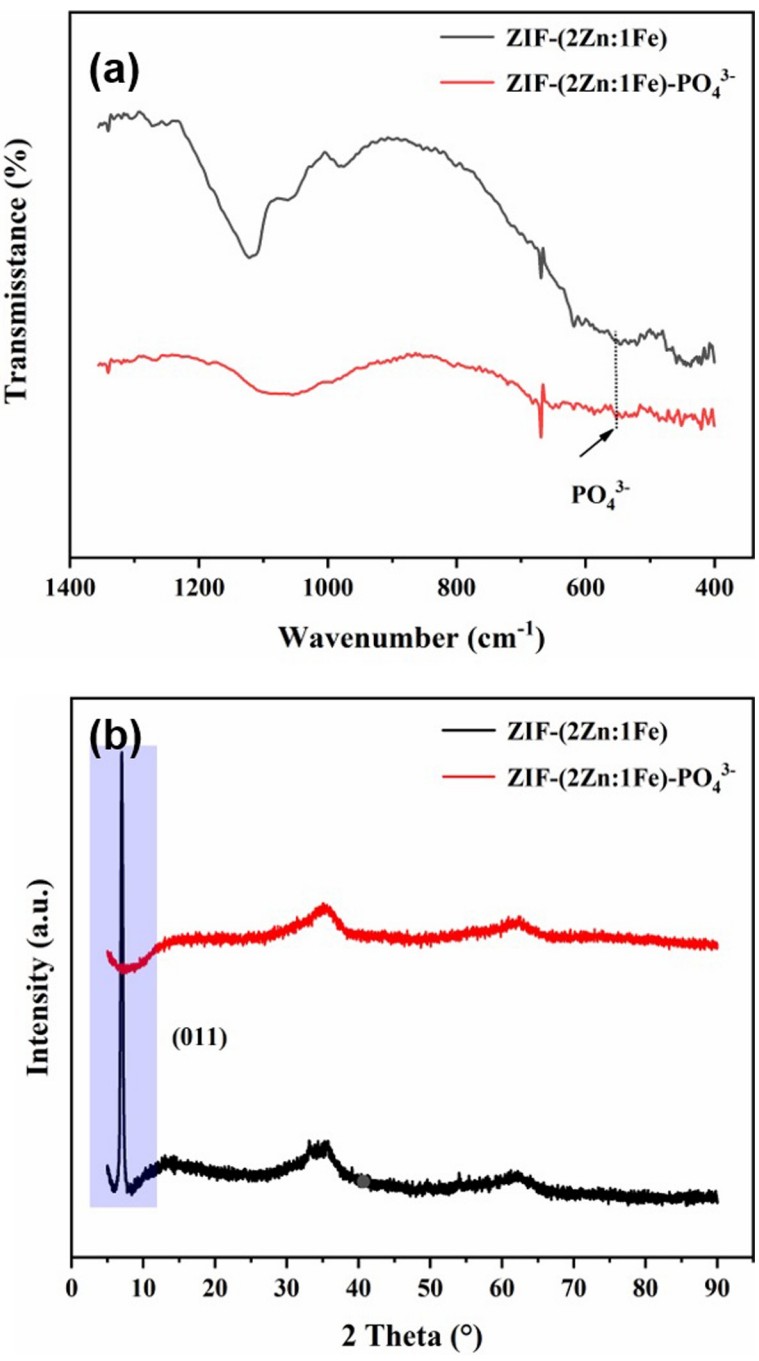

**Fig 10. FT-IR spectra and XRD patterns of ZIF-(2Zn:1Fe) before and after use.**

As shown in Fig 11A and 11B, both full-scan and high-resolution XPS spectra showed clear P signal, which confirm that the adsorption reaction occurred. These results agreed with the EDX results. In the high-resolution Fe 2p XPS spectra, the typical binding energy peaks of 711.00, 718.50 and 724.63 eV after adsorption jumped to 711.10, 718.60, and 724.64 eV, respectively. Similarly, in the Zn 2p XPS spectra, the typical peaks at binding energies of 1021.37 and 1044.47 eV were shifted to 1021.45 and 1044.50 eV. These data could demonstrate

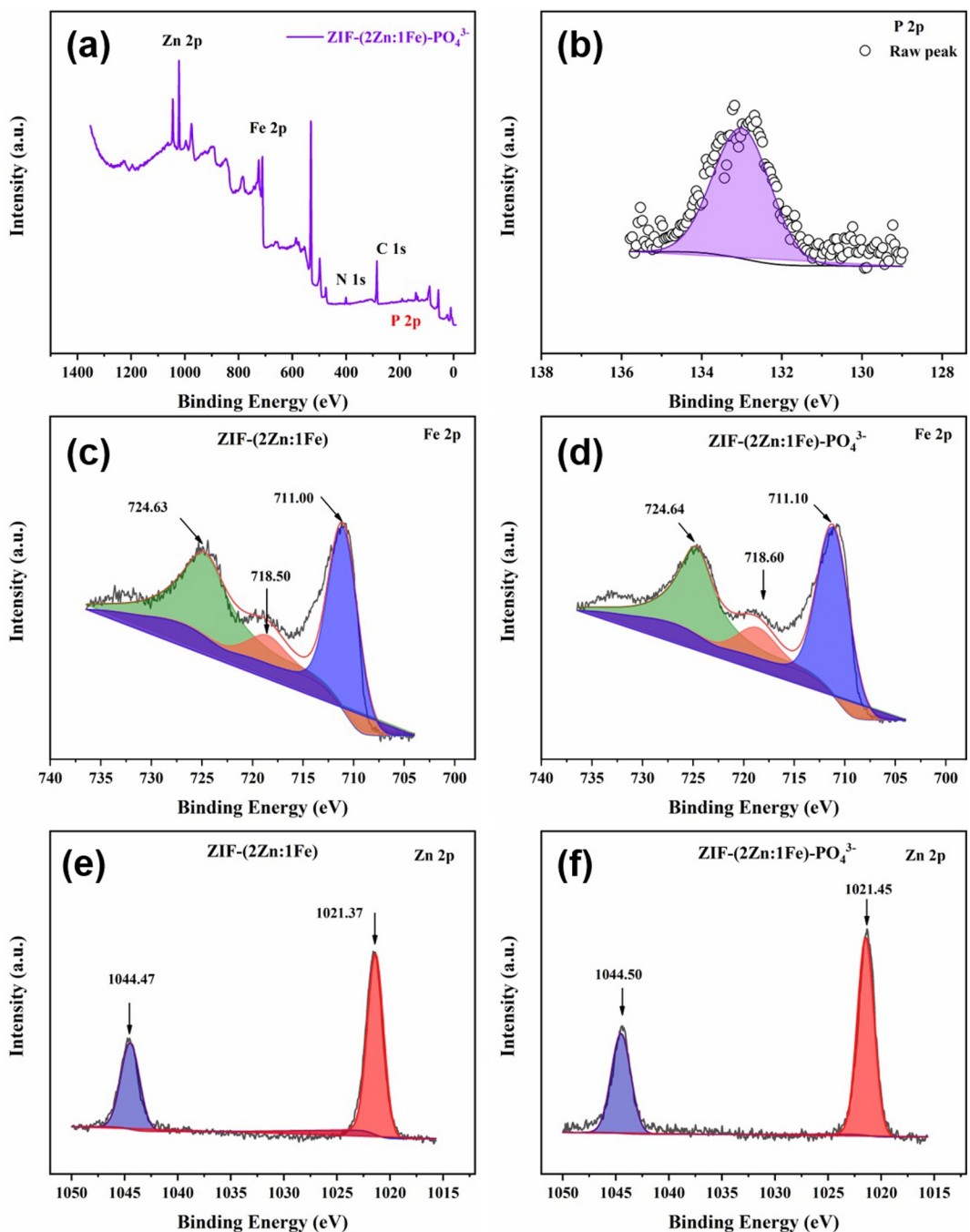

**Fig 11.** Full XPS spectrum (a), P 2p XPS spectrum of ZIF-(2Zn:1Fe)- $PO_4^{3-}$ (b); Fe 2p XPS spectra (c, d) and Zn 2p XPS spectra (e, f) of ZIF-(2Zn:1Fe) and ZIF-(2Zn:1Fe)- $PO_4^{3-}$.

that both Zn-based and Fe-based groups were active sites for the adsorption of phosphate, while the introduction of Fe atoms created new adsorption sites [29, 42–45].

## 4. Conclusions

In conclusion, we have presented a novel Fe-doped ZIF-8 frameworks, synthesized by solvothermal reactions and used as a superior adsorbent for phosphate removal from water. Based on the different characterization results, we found that Fe atoms were successfully doped into the ZIF-8 framework, and the as-prepared Fe-doped ZIF-8 (denoted as ZIF-(2Zn:1Fe)) exhibited a polyhedral morphology with a large specific surface area of 157.64 $m^2$/g and a pore size of 3.055 nm. The adsorption results showed that the molar ratio of Fe and Zn in the precursors had significant effects on the adsorption performance. The adsorption kinetics conformed to a pseudo-second-order model with a high coefficient of determination ($R^2$) of 0.9983. The adsorption isotherms matched the Langmuir model ($R^2$ = 0.9994) better than the Freundlich model ($R^2$ = 0.7501). This result suggested that the adsorption of phosphate by ZIF-(2Zn:1Fe) was chemisorbed on a homogeneous surface. Under this condition, the maximum adsorption amount could reach 38.24 mg/g. An acidic environment was beneficial for the adsorption reaction. The inhibition of common anions was in the order of $NO_3^- > CO_3^{2-} > SO_4^{2-} > Cl^-$. Compared to the influence of pH and coexisting ions, the effect of ionic strength on adsorption was negligible. Based on the comprehensive characterization results, it is the complex interactions within the surface spheres that mainly influence the adsorption process. This metal-doping strategy may be effective for the synthesis of other enhanced MOF nanostructures.

## Supporting information

**S1 File.**
(DOCX)

## Author Contributions

**Conceptualization:** Zhijia Miao.

**Data curation:** Zhijia Miao.

**Formal analysis:** Xueqiang Song.

**Funding acquisition:** Zhijia Miao.

**Investigation:** Shuoyang Li.

**Project administration:** Xueqiang Song, Zhen Jiao.

**Resources:** Xiaolei Wang.

**Software:** Xiaolei Wang.

**Supervision:** Zhen Jiao.

**Validation:** Hao Wang.

**Visualization:** Hao Wang.

**Writing – original draft:** Zhijia Miao, Shuoyang Li.

**Writing – review & editing:** Zhen Jiao.

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
