## [Decision Letter · Decision Letter 0]

5 Aug 2024

PONE-D-24-20593Facile synthesis of Fe-doped ZIF-8 and its adsorption of phosphate from water: Performance and mechanismPLOS ONE

Dear Dr. Jiao,

Thank you for submitting your manuscript to PLOS ONE. After careful consideration, we feel that it has merit but does not fully meet PLOS ONE’s publication criteria as it currently stands. Therefore, we invite you to submit a revised version of the manuscript that addresses the points raised during the review process.

We look forward to receiving your revised manuscript.

Kind regards,

Sheng Yu

Academic Editor

PLOS ONE

Fabrication of Ce-doped MIL-100(Fe), its adsorption performance, and the mechanisms to adsorb phosphate from water - https://doi.org/10.1016/j.eti.2022.102847

(Among others)

In your revision ensure you cite all your sources (including your own works), and quote or rephrase any duplicated text outside the methods section. Further consideration is dependent on these concerns being addressed.

4. PLOS requires an ORCID iD for the corresponding author in Editorial Manager on papers submitted after December 6th, 2016. Please ensure that you have an ORCID iD and that it is validated in Editorial Manager. To do this, go to ‘Update my Information’ (in the upper left-hand corner of the main menu), and click on the Fetch/Validate link next to the ORCID field. This will take you to the ORCID site and allow you to create a new iD or authenticate a pre-existing iD in Editorial Manager. Please see the following video for instructions on linking an ORCID iD to your Editorial Manager account: https://www.youtube.com/watch?v=_xcclfuvtxQ.

5. We note that Figures 3 and 9 in your submission contain [map/satellite] images which may be copyrighted. All PLOS content is published under the Creative Commons Attribution License (CC BY 4.0), which means that the manuscript, images, and Supporting Information files will be freely available online, and any third party is permitted to access, download, copy, distribute, and use these materials in any way, even commercially, with proper attribution. For these reasons, we cannot publish previously copyrighted maps or satellite images created using proprietary data, such as Google software (Google Maps, Street View, and Earth). For more information, see our copyright guidelines: http://journals.plos.org/plosone/s/licenses-and-copyright.

1. You may seek permission from the original copyright holder of Figures 3 and 9 to publish the content specifically under the CC BY 4.0 license.  

Reviewers' comments:

Reviewer's Responses to Questions

**Comments to the Author**

1. Is the manuscript technically sound, and do the data support the conclusions?

Reviewer #1: Yes

Reviewer #2: Yes

2. Has the statistical analysis been performed appropriately and rigorously? 

Reviewer #1: Yes

Reviewer #2: Yes

3. Have the authors made all data underlying the findings in their manuscript fully available?

Reviewer #1: Yes

Reviewer #2: Yes

4. Is the manuscript presented in an intelligible fashion and written in standard English?

Reviewer #1: Yes

Reviewer #2: Yes

5. Review Comments to the Author

Reviewer #1: The manuscript is exceptionally well-written, demonstrating clear logical progression and comprehensive characterization. The analysis seamlessly transitions from the adsorption mechanism to the adsorption effect, with an in-depth examination throughout. The section on material characterization is particularly commendable, clearly elucidating the impact of iron addition on the Zeolitic Imidazolate Framework (ZIF) structure and the underlying adsorption principles. The X-ray Photoelectron Spectroscopy (XPS) comparisons pre- and post-adsorption effectively illustrate the adsorption capabilities of the material.

Including adsorption kinetics further enriches the study, offering a well-rounded evaluation of the material's performance. I recommend publication without further amendments. However, for future studies, it would be advantageous to explore the adsorption's durability and the material's reusability in similar applications. Despite this, the current focus on the adsorption performance and the influence of iron on ZIF is addressed thoroughly and executed flawlessly.

Some relevant reference should be cited:

Removal of Cr (VI) from aqueous solutions using polymer nanotubes. Journal of Materials Science 55.1 (2020): 163-176.

Efficient removal of Cr (VI) and Pb (II) from aqueous solution by magnetic nitrogen-doped carbon. Frontiers of Chemical Science and Engineering 15 (2021): 1185-1196.

Reviewer #2: This study prepared ZIF-8 w to remove phosphate from water. To help the authors improve this manuscript, specific suggestions and comments are put forward as follows:

1. Please check typo word.

2. PO43- is phosphoric acid or phosphate? Please make sure again which target for this study.

3. Do you have reference for this 606 cm1 which was attributed to the stretching vibration of Fe-N? I think Fe-O, but please make sure again.

4. Please add reference in all FTIR data.

5. These observations indicated that small amounts of oxides or hydroxides may be present in the iron-containing products (Page 9). How do you measure oxides and hydroxides through sem analysis?

6. Can you write details how to measure phosphate? This technique can measuring until 1 g/L?

6. PLOS authors have the option to publish the peer review history of their article (what does this mean?). If published, this will include your full peer review and any attached files.

Reviewer #1: No

Reviewer #2: No

---

## [Author Response · Author response to Decision Letter 0]

4 Sep 2024

Dear editor,

Thank you very much for your and the reviewers’ comments, which are very precious and valuable for us to revise our manuscript (PONE-D-24-20593). We have revised the MS carefully according to the comments. The revision parts were highlighted in yellow background in the revised manuscript. The point-to-point responses to the comments are listed as follows:

Editor:

Response: 

Thanks very much for your careful examination. We have revised our manuscript according the PLOS ONE requirements. 

Fabrication of Ce-doped MIL-100(Fe), its adsorption performance, and the mechanisms to adsorb phosphate from water - https://doi.org/10.1016/j.eti.2022.102847

In your revision ensure you cite all your sources (including your own works), and quote or rephrase any duplicated text outside the methods section. Further consideration is dependent on these concerns being addressed.

Response: Thank you very much. We carefully revised the overlapping contents, and these were highlighted in yellow background. We added several references into the reference list.

Response: Thank you very much. We have provided a complete Data Availability Statement in the submission form.

4. PLOS requires an ORCID iD for the corresponding author in Editorial Manager on papers submitted after December 6th, 2016. Please ensure that you have an ORCID iD and that it is validated in Editorial Manager. To do this, go to ‘Update my Information’ (in the upper left-hand corner of the main menu), and click on the Fetch/Validate link next to the ORCID field. This will take you to the ORCID site and allow you to create a new iD or authenticate a pre-existing iD in Editorial Manager. Please see the following video for instructions on linking an ORCID iD to your Editorial Manager account: https://www.youtube.com/watch?v=_xcclfuvtxQ.

Response: Thank you very much.

5. We note that Figures 3 and 9 in your submission contain [map/satellite] images which may be copyrighted. All PLOS content is published under the Creative Commons Attribution License (CC BY 4.0), which means that the manuscript, images, and Supporting Information files will be freely available online, and any third party is permitted to access, download, copy, distribute, and use these materials in any way, even commercially, with proper attribution. For these reasons, we cannot publish previously copyrighted maps or satellite images created using proprietary data, such as Google software (Google Maps, Street View, and Earth). For

Response: Thank you very much. The Figures 3 and 9 is the SEM images, in which some analysis information are included. And they are not copyrighted.

Reviewer #1: 

The manuscript is exceptionally well-written, demonstrating clear logical progression and comprehensive characterization. The analysis seamlessly transitions from the adsorption mechanism to the adsorption effect, with an in-depth examination throughout. The section on material characterization is particularly commendable, clearly elucidating the impact of iron addition on the Zeolitic Imidazolate Framework (ZIF) structure and the underlying adsorption principles. The X-ray Photoelectron Spectroscopy (XPS) comparisons pre- and post-adsorption effectively illustrate the adsorption capabilities of the material.

Including adsorption kinetics further enriches the study, offering a well-rounded evaluation of the material's performance. I recommend publication without further amendments. However, for future studies, it would be advantageous to explore the adsorption's durability and the material's reusability in similar applications. Despite this, the current focus on the adsorption performance and the influence of iron on ZIF is addressed thoroughly and executed flawlessly.

Some relevant reference should be cited:

Removal of Cr (VI) from aqueous solutions using polymer nanotubes. Journal of Materials Science 55.1 (2020): 163-176.

Efficient removal of Cr (VI) and Pb (II) from aqueous solution by magnetic nitrogen-doped carbon. Frontiers of Chemical Science and Engineering 15 (2021): 1185-1196.

Response: 

Thank you very much for your careful review and giving us the opportunity to revise our paper. We have revised the paper carefully according to the comments. I have meticulously read the two references you suggested and have inserted them into the article for those reading it as a reference.

Reviewer #2: 

This study prepared ZIF-8 w to remove phosphate from water. To help the authors improve this manuscript, specific suggestions and comments are put forward as follows:

1. Please check typo word.

Response: Thank you very much. We have meticulously checked the entire text and corrected some typo words.

2. PO43- is phosphoric acid or phosphate? Please make sure again which target for this study.

Response: Thank you very much. In the experiment, we dissolved potassium dihydrogen phosphate (KH2PO4) into ultrapure water to achieve a stock solution. The phosphate is viewed as the target pollutant in this study. The phosphate may exist as PO43-, HPO42-, H2PO4- and H3PO4 in solution, whose proportion and transformation depends on the solution pH.

3. Do you have reference for this 606 cm-1 which was attributed to the stretching vibration of Fe-N? I think Fe-O, but please make sure again.

Response: Thank you for your comment. By reading references, we found that this 606 cm-1 band belongs to F the stretching vibration of Fe-O. The reference is listed as follows:

Jirásek J, Čejka J, Vrtiška L, Matýsek D, Ruan X, Frost RL. Molecular structure of the phosphate mineral koninckite - a vibrational spectroscopic study. Journal of Geosciences. 2018: 271-9. doi: 10.3190/jgeosci.243

4. Please add reference in all FTIR data.

Response: Thank you very much. The reference is seen below: 

Wu C, Xie D, Mei Y, Xiu Z, Poduska KM, Li D, et al. Unveiling the thermolysis natures of ZIF-8 and ZIF-67 by employing in situ structural characterization studies. Physical Chemistry Chemical Physics. 2019; 21(32): 17571-7. doi: 10.1039/c9cp02582k

5. These observations indicated that small amounts of oxides or hydroxides may be present in the iron-containing products (Page 9). How do you measure oxides and hydroxides through SEM analysis?

Response: Thanks for your valuable comment. SEM could suggest the microstructure and morphology of the material. In analysis section, we found there were many small particles on the surface of Fe-doped ZIF-8 whereas no similar phenomenon was for ZIF-8. Combining XRD results, we supposed that these particles may be the newly formed oxides or hydroxides. 

6. Can you write details how to measure phosphate? This technique can measure until 1 g/L?

Response: In our experiment, all phosphate concentrations were analyzed by ammonium molybdate spectrophotometric at a wavelength of 700 nm on a UV-1780 spectrophotometer. This method could provide a mg/L (PO43-) level of determination. In the manuscript, we mentioned 1g/L of phosphate. It was a stock solution, prepared by dissolving a predetermined amount of KH2PO4 in ultrapure water. This is a calculated concentration value not a measuring value by a UV-1780 spectrophotometer. When analyzing high concentration of phosphate, a diluting program is usually considered as an essential treatment.

---

## [Decision Letter · Decision Letter 1]

17 Sep 2024

Facile synthesis of Fe-doped ZIF-8 and its adsorption of phosphate from water: Performance and mechanism

PONE-D-24-20593R1

Dear Dr.Zhen Jiao,

We’re pleased to inform you that your manuscript has been judged scientifically suitable for publication and will be formally accepted for publication once it meets all outstanding technical requirements.

Kind regards,

Sheng Yu

Academic Editor

PLOS ONE

Additional Editor Comments (optional):

Reviewers' comments:

Reviewer's Responses to Questions

**Comments to the Author**

1. If the authors have adequately addressed your comments raised in a previous round of review and you feel that this manuscript is now acceptable for publication, you may indicate that here to bypass the “Comments to the Author” section, enter your conflict of interest statement in the “Confidential to Editor” section, and submit your "Accept" recommendation.

Reviewer #2: All comments have been addressed

2. Is the manuscript technically sound, and do the data support the conclusions?

Reviewer #2: Yes

3. Has the statistical analysis been performed appropriately and rigorously? 

Reviewer #2: I Don't Know

4. Have the authors made all data underlying the findings in their manuscript fully available?

Reviewer #2: Yes

5. Is the manuscript presented in an intelligible fashion and written in standard English?

Reviewer #2: Yes

6. Review Comments to the Author

Reviewer #2: I think the manuscripts well. But, my questions is for FTIR spectra? Why only for 1400 cm-1, usually until 4000 cm (wavenumbers). Please revise the FTIR spectra. I think O-H group also influence for adsorption process.

7. PLOS authors have the option to publish the peer review history of their article (what does this mean?). If published, this will include your full peer review and any attached files.

Reviewer #2: No

---

## [Editor Report · Acceptance letter]

1 Oct 2024

PONE-D-24-20593R1 

PLOS ONE

Dear Dr. Jiao, 

I'm pleased to inform you that your manuscript has been deemed suitable for publication in PLOS ONE. Congratulations! Your manuscript is now being handed over to our production team.

Kind regards, 

on behalf of

Dr. Sheng Yu 

Academic Editor

PLOS ONE